# Influence of the Nutritional Composition of Quinoa (*Chenopodium quinoa* Willd.) on the Sensory Quality of Cooked Quinoa

**DOI:** 10.3390/foods14060988

**Published:** 2025-03-14

**Authors:** Shengyuan Guo, Chaofan Zhao, Jiankang Zhou, Zhuo Zhang, Wenting Wang, Yuting Zhu, Chuan Dong, Guixing Ren

**Affiliations:** 1Environmental Science Research Institute, Shanxi University, Taiyuan 030006, China; guoshengyuan217@163.com; 2School of Life Science, Shanxi University, Taiyuan 030006, China; 202223110045@email.sxu.edu.cn (C.Z.); jasminzhou622@163.com (J.Z.); 202213101003@email.sxu.edu.cn (Z.Z.); xtt11082021@163.com (W.W.); 202423117016@email.sxu.edu.cn (Y.Z.)

**Keywords:** quinoa, nutrients, sensory evaluation, electronic tongue

## Abstract

In order to explore the effect of the content of nutritional components of quinoa on its sensory quality, 22 quinoa varieties were collected from 11 major quinoa-producing areas at home and abroad as experimental materials. The contents of total starch, protein, fat, crude fiber, ash, VB1, VB2, moisture and saponin were determined, and the sensory evaluation and electronic tongue analysis of cooked quinoa were carried out. The sensory quality of quinoa was comprehensively evaluated by correlation analysis, principal component analysis and cluster analysis. The results show that the contents of various nutritional components had significant effects on the sensory quality of quinoa (*p* < 0.05). Quinoa with high starch, high VB1, moderate fat, moderate moisture, low protein, low crude fiber, low ash and low saponin content had better cooking quality and was more popular. Electronic tongue analysis showed that the sweet response value of cooked quinoa was the highest, followed by the bitter response value. No. 12 and No. 9 cooked quinoa samples had the best comprehensive taste, the highest sensory score and the best cooking quality. This study preliminarily clarified the relationship between the content of different nutritional components in quinoa and its sensory quality, which could provide reference for the selection of raw materials and breeding of quinoa varieties for different processing purposes.

## 1. Introduction

Quinoa (*Chenopodium quinoa* Willd.) is an annual dicotyledonous plant of the genus *Chenopodium* in *Amaranthaceae*, which is native to the Andean region of South America and is an agricultural crop similar to grain. About 7000 years ago, quinoa was domesticated and cultivated by local residents as a major food crop [1]. At the present time, the main global producers of quinoa are Peru, Bolivia and Ecuador, which account for more than 90% of total global production [2]. Additionally, more and more places such as the USA, Europe, Canada, Australia, China and India also cultivate it [3]. Quinoa is rich in starch, lipids, protein, dietary fiber and nutrients that meet the body’s daily intake [4]. Although quinoa has high cultivation worth, nutritional worth and functional worth, it has been ignored for thousands of years. Until the past 50 years, with the rapid growth of the global population and the increasingly severe natural environment and agricultural production conditions, it is urgent to find a high-quality food to meet people’s needs for daily consumption [5]. In 2013, quinoa was officially recommended by FAO as the perfect “full nutrition food” for human beings. Quinoa is used in the food industry in a wide variety of products, including noodles, vegetable soups, pasta and more [6]. Quinoa rice is made from quinoa after processing and shelling. Its cooking and eating method is similar to that of rice, and it can be cooked into cooked quinoa for eating.

The sensory quality of cereals is closely related to their main nutritional components. The higher the starch content of rice and brown rice, the greater the final viscosity and retrogradation value of gelatinization [7]. The starch content of millet is positively correlated with viscosity, hardness, elasticity and other indicators. In addition, it is also significantly correlated with sensory indicators such as appearance and palatability of millet porridge [8]. Rice with low protein content has better palatability and taste, and has a significant correlation with the elasticity and viscosity of rice [9]. Protein and fat contents of millet porridge also have significant correlation with sensory quality [10]. The fat content of rice and millet has a significant correlation with hardness [11]. The fat in rice with amylose reduce the quality of rice and rice flour [12].

The contents and compositions of starch, protein and fat in quinoa are quite different from those in rice and millet. The starch content of quinoa is 53.5~69.2% [13], which is lower than that of rice 57.15~80% [14] and millet 70.9~82.7% [15]. The protein content of quinoa is 12~22% [16,17], which is significantly higher than that of rice 6.18~8.98% and millet 10.09~14.35% [15]. The fat content of quinoa is 4.4~8.8% [18], which is higher than that of rice 0.27~4.54% and millet 2.8~6.6%. The content of saponins in quinoa varies depending on the variety and site, with 27.26 mg/g of saponins in quinoa seeds [19]. The nutritional content of different quinoa germplasms also has great differences, so the edible quality of cooked quinoa made from quinoa with different nutritional contents may also be different. Some researchers reported that the higher the protein content of quinoa, the greater the viscosity of quinoa rice [20], which is completely different from Li et al., who reported that the higher the protein content of rice, the worse the viscosity of rice [14]. The presence of saponins may negatively affect the overall flavor of quinoa. For example, in the brewing of quinoa beer, saponins and metal ions can make the flavor bitter and cause turbidity [21].

The cooking quality of cooked quinoa can be evaluated by artificial sensory and analyzed by an electronic tongue instrument. Wu et al. took 21 quinoa germplasms as research materials, referred to the rice sensory evaluation standard, combined with the characteristics of quinoa, and took texture, odor, taste and color as the main indicators, establishing the sensory evaluation system of quinoa [22]. It is pointed out that there are differences in the odor of different varieties of quinoa after steaming, such as burnt aroma, nut aroma, grass aroma, earthy flavor and woody flavor. At the same time, quinoa also has sweet, bitter, grain flavor, bean flavor and nut flavor. Yueyuan et al. studied the effects of soaking time and cooking time on the sensory quality of cooked quinoa [23]. The results showed that the total score of sensory evaluation was the highest when soaking time was 40~80 min and cooking time was 30 min.

At present, the research on the relationship between nutritional components and edible quality of quinoa is limited, and the impact of multiple nutritional components has not been comprehensively evaluated. In addition, there are relatively few reports on the effects of nutritional components on the sensory quality of cooked quinoa. Therefore, this study intends to systematically analyze the correlation between the content of nutritional components of quinoa and edible quality by comprehensively determining the content of nutritional components of different varieties of quinoa, and conducting artificial sensory evaluation and electronic tongue analysis of cooked quinoa, so as to identify the nutritional components that have the most significant impact on the edible quality of cooked quinoa, in order to provide theoretical support for quinoa food processing and breeding of high-quality varieties.

## 2. Materials and Methods

### 2.1. Materials

Twenty-two quinoa varieties were collected from eleven major quinoa-producing regions in China and abroad for this study, as shown in Table 1. The threshed and dried quinoa was stored in a refrigerator at 4 °C and removed for hulling before the experiment. Analytical grade sodium acetate anhydrous, papain (enzyme activity ≥ 800 U/mg) and amylase (enzyme activity ≥ 3700 U/g) were purchased from Aladdin Biochemical Technology Co., Ltd. (Shanghai, China). Potassium ferricyanide, analytical grade, was purchased from Tianjin Guangfu Science and Technology Development Co., Ltd. (Tianjin, China). Analytical grade magnesium acetate was purchased from Sinopharm Chemical Reagent Co., Ltd. (Beijing, China). Sodium acetate of analytical grade was purchased from Sigma Aldrich Trading Co., Ltd. (Beijing, China).

### 2.2. Determination of Nutrient Content

The approximate parameters of moisture, ash, protein, fat, fiber, vitamin B1 (VB1) and vitamin B2 (VB2) were determined by the methods of GB 5009.3-2016, 5009.4-2016, 5009.5-2016, 5009.6-2016, 5009.10-2003, GB 5009.84-2016, 5009.85-2016, respectively [24,25,26,27,28,29,30]. The total starch content was determined by the total starch determination kit (Megazyme International Ireland Ltd., Wicklow, Ireland). The total saponin content of quinoa was determined by vanillin-perchloric acid colorimetric method with reference to the method of Xiong et al. [31].

### 2.3. Preparation of Cooked Quinoa

In the preparation of cooked quinoa, 30 g quinoa was accurately weighed. Subsequently, the quinoa must undergo a rigorous washing process, to be performed thrice in potable water, with the objective of eliminating any residual saponin, identifiable by the absence of a white froth. Following this, the quinoa is to be submerged in potable water for a duration of 30 min, facilitating the complete hydration of the grains, which is essential for the subsequent gelatinization process during cooking. The hydrated quinoa is then transferred to a ceramic vessel and subjected to steaming for a period of 40 min, yielding fresh cooked quinoa.

### 2.4. Sensory Evaluation of Cooked Quinoa

The evaluation team was composed of 18 researchers who had been trained in sensory evaluation. The sensory score of cooked quinoa was slightly modified according to the previous method [32]. The evaluation indexes included chrominance, structure, flavor, hardness, chewiness, viscosity and taste (Appendix A). The average score of the team members was calculated as the evaluation result.

### 2.5. Electronic Tongue Detection

The SA402b electronic tongue system (INSENT Company, Atsugi, Japan) was used to analyze taste differences. This system contained six test sensors, among which CA0, GL1, C00, CT0, AAE and AE1, which are sensitive to sourness, sweetness, bitterness, saltiness, umami and astringency. The precise measurement of 50 g cooked quinoa was to be undertaken, followed by the preparation of a homogenate by blending it with water at a material-to-liquid ratio of 1:5 (*w*/*w*). The homogenate was then subjected to centrifugal separation at 8000 r/min for a duration of 10 min. Sixty milliliters of the supernatant was transferred into a beaker for subsequent analysis using an electronic tongue. Each sample was repeated 5 times, and the data of the last 3 times were selected as the original data for analysis.

### 2.6. Statistical Analysis

The experiment was repeated three times, and the results were expressed as mean ± standard deviation. Statistical analysis was performed using SPSS version 26.0. The data were subjected to one-way analysis of variance (ANOVA) to determine the differences between samples. Pearson correlation analysis, principal component analysis (PCA) and cluster analysis were performed using Origin Software (Origin 2022; Origin Lab, Northampton, MA, USA).

## 3. Results and Discussion

### 3.1. Nutrients Content

The results of the determination of the content of the nutritional components of the 22 quinoa samples, all on a dry weight basis, are shown in Table 2. Starch was the most abundant component in quinoa, accounting for about 42.47–64.03% of the dry weight of quinoa. The highest total starch content was found in sample No. 13 and the lowest in sample No. 11. The starch content of the main quinoa varieties in different regions ranged from 571.6 to 610.5 g/kg [33]. Different regions have diversified dimensions and altitudes, resulting in variation in starch content in quinoa seeds produce in different regions [34]. Starch content is one of the most important factors in determining the eating quality of cooked quinoa. Wu et al., studied the textural properties of different varieties of quinoa and found that the content of straight-chain amylopectin [20], which separates the starch, showed a high positive correlation with the hardness, stickiness, cohesion, gelatinization and chewiness. However, the effect of total amylose content of quinoa on its sensory quality has not been reported.

In this study, the protein content of 22 quinoa species ranged from 8.99% to 16.79%. Among them, samples No. 5, No. 7, No. 11 and No. 13 had relatively high protein content. The average protein content of quinoa was higher than that of traditional crops such as barley, corn and rice, and close to that of wheat [35]. Significant differences in protein content between different quinoa varieties may be influenced by endogenous factors (e.g., genetic differences) and exogenous factors (e.g., cultivation conditions, storage forms). The good stability of quinoa protein is based on the existence of a large number of disulfide bonds in its spatial structure, so quinoa protein is not easily affected by the processing conditions, and can well maintain its original nutritional properties and functional properties [36]. Protein is the second most important nutrient in quinoa after starch. Wu et al. showed that the protein content of different varieties of quinoa had a significant positive correlation with the hardness, chewiness, cohesion and viscosity, which suggests that the higher the protein content of quinoa, the harder the cooked quinoa may appear [20]. Meng et al. showed that the sensory quality of pork patties supplemented with quinoa protein was consistently better than that of the control group throughout the freeze–thaw cycle of pork patties [37]. However, the effect of quinoa protein content on the sensory quality of quinoa rice has not yet been reported and further studies are needed.

The fat content of 22 quinoa species used in this study ranged from 2.94% to 8.88%. Among them, the fat content of sample No. 21 was the lowest (2.94%), which was significantly lower than that of other quinoa varieties. The coefficient of variation for fat in quinoa samples was larger relative to starch and protein, suggesting that fat content has a greater influence on quinoa quality. The fat content in quinoa varies with different varieties, environment, treatment methods and other factors. In addition to protein and starch, fat content can also affect food quality. Li et al.’s research on the edible quality of different varieties of rice showed that the higher the fat content of rice, the smaller the peak viscosity of gelatinization, and the smaller the hardness, cohesion, resilience, adhesion and chewiness of rice [14]. Zhen et al.’s research on the eating quality of rice shows that the higher the fat content, the more complete the appearance structure of rice, the better the taste and the higher the sensory score and taste value [38]. However, the relationship between fat content and sensory quality of quinoa needs to be studied.

The crude fiber content of the 22 quinoa species used in this study ranged from 0.76% to 8.67%, with the highest coefficient of variation (69.5) compared to the other metrics. Quinoa dietary fiber is highly absorbent and increases in volume by 3–4 times after cooking, which enhances satiety [39]. However, quinoa’s rich dietary fiber can disrupt the gluten network structure. When making pasta, such as noodles or bread, quinoa dough has poor viscoelasticity, breaks easily, has a high solubility rate and lacks good elasticity and ductility [40]. Quinoa is rich in water-soluble B vitamins and is a good natural source of vitamins, meeting 40% and more of the adult daily requirement [41]. Although vitamin B itself has a limited effect on taste, it may be involved in the formation of flavor substances during cooking, thus indirectly affecting the quinoa eating experience [42]. The ash content of different quinoa varieties in this study also varies greatly, and its content is inextricably linked to the soil environment, and is also related to climate, variety and growth period, such as quinoa grown in a soil environment rich in minerals such as Se and Mg, and its Se and Mg content is also higher [43]. During cooking, moisture enters the rice grain through the cellular interstices to make starch pasty, so the moisture status also has an important effect on the eating quality [10].

The saponin content of the 22 quinoa samples in this study ranged from 0.14% to 0.31%. Among them, samples No. 2 and No. 20 had the lowest saponin content, while sample No. 4 had the highest saponin content. According to the saponin content, quinoa is divided into sweet quinoa (saponin content < 0.11%) and bitter quinoa (saponin content > 0.11%) [44]. There is a wide variety of saponins, mainly including triterpenoid saponins, which are the main source of quinoa’s bitter flavor [45]. Bitterness and anti-nutritional properties of saponins are the main factors affecting the acceptability of quinoa. In order to improve the eating quality of quinoa, researchers have developed several saponin removal methods, including water washing, enzymatic digestion and alkali treatment [46]. These methods can effectively reduce the saponin content and thus alleviate the bitter flavor.

### 3.2. Sensory Evaluation

Sensory evaluation is the most intuitive index to describe and judge food quality through human taste, touch, vision, smell and hearing. The sensory evaluation scores of 22 cooked quinoa varieties are shown in Figure 1A, and the total sensory evaluation scores showed significant differences. The coefficient of variation of chrominance, structure, flavor, hardness, chewiness, viscosity and taste in sensory evaluation scores of cooked quinoa were 8.07%, 6.84%, 5.90%, 7.53%, 7.50%, 5.38% and 12.93%, respectively, and the coefficient of variation of taste was greater than 10%, indicating that there were significant differences in taste among 22 cooked quinoa samples. The total score of sensory evaluation of cooked quinoa samples ranged from 64.4 to 82.3, and the scores of all sensory evaluation indexes of sample No. 12 were higher, especially the taste, flavor and viscosity. The evaluators generally believed that No. 12 cooked quinoa sample tasted soft and sticky, and had a nutty flavor when chewing. Soaking can soften the quinoa and reduce cooking time for a softer texture [47]. The chrominance, structure and chewiness score of No. 9 cooked quinoa sample were relatively high. Most evaluators believed that its aroma was rich and chewy, and the grain structure was relatively complete. No. 1 and No. 2 samples were highly praised by evaluators in taste, viscosity and hardness, so the total score of sensory evaluation was high. The sensory index scores of No. 4, No. 6 and No. 18 samples were low, the taste was poor, and there was no viscosity, so they were not suitable for making cooked quinoa. In addition to varying due to the type of quinoa (e.g., white, red, black), the degree of cleaning of the quinoa, the way it is cooked and the steaming time can make a big difference in the texture of the quinoa rice [48].

### 3.3. Electronic Tongue Analysis

The electronic tongue system consists of various taste sensors that are capable of detecting taste compounds in food samples [49]. As shown in Figure 1B, the response values of six sensors are displayed, including those of the sourness, sweetness, bitterness, umami, freshness and astringency sensors. There was a significant difference (*p* < 0.05) in the electronic tongue evaluation values of 22 cooked quinoa. Overall, cooked quinoa had relatively the highest sweetness response value, followed by the bitter response value. Among them, cooked quinoa No. 19, No. 20 and No. 22 had higher sweetness and relatively lower saponin content. The main carbohydrate of quinoa is starch, and during the cooking process, the starch grains will absorb water and swell, dissolve and partially hydrolyze into soluble sugars, such as glucose and maltose, which give quinoa a certain sweetness [50]. Saponins are the main anti-nutrients in quinoa, which will affect the edible taste of quinoa and produce bitter and astringent taste [51]. Most of the saponins could be removed by washing or soaking well before cooking. The umami of cooked quinoa was also evident in the fact that the proteins in quinoa were partially hydrolyzed, releasing amino acids, especially the increase in umami amino acids such as glutamic acid, which significantly enhances the umami of the taste [52]. Cooked quinoa had the lowest sourness value, indicating that the flavor of cooked quinoa was in line with people’s taste and suitable for consumption. During cooking, the organic acids in quinoa may be partially broken down or converted, thus reducing the acidic flavor [52]. As shown in Figure 1C, specifically, cooked quinoa sample No. 22 had the highest sweetness value, but its bitter and astringent values were also relatively high, resulting in a relatively low sensory score. Similarly, cooked quinoa sample No. 7 had the highest umami and saltiness values, but also had relatively high bitterness and astringency values, which when combined, resulted in a moderate sensory score. Cooked quinoa samples Nos. 12 and 9 had relatively high sweetness and umami values, which far outweighed the effect of their lower bitterness and astringency, resulting in the highest sensory scores. Therefore, in terms of organoleptic qualities, quinoa samples No. 12 and No. 9 were more suitable for cooking and eating as cooked quinoa.

### 3.4. Comprehensive Evaluation

#### 3.4.1. Correlation Analysis

The correlation analysis between the content of different nutritional fractions of quinoa and the scores of sensory evaluation indexes of cooked quinoa is shown in Figure 2. The total protein content was highly significantly negatively correlated with the chrominance score (*p* < 0.01) and also showed negative correlation with the total sensory evaluation score. It has been shown that as the protein content of rice increases, it usually deteriorates the cooking quality and palatability of rice [53]. Li et al. also reported that the protein content of rice was significantly negatively correlated with the sensory score of rice [9]. Proteins can compete with starch molecules for free water, resulting in insufficient starch water absorption and affecting its degree of pasting, which in turn leads to poor cooking sensory quality [54]. Total starch content was highly significantly and positively correlated with chrominance, flavor, taste and total score of sensory evaluation (*p* < 0.01), and also significantly correlated with hardness (*p* < 0.05). During sensory evaluation of chewing the rice grains, the salivary glands in the mouth secrete a variety of enzymes such as α-amylase and glycosidase, which metabolize the starch in the quinoa rice into soluble sugars, resulting in a sweet taste [55]. Therefore, the higher the total amylose content, the more favorable the formation of sweetness in cooked quinoa, which improves its taste score. The study of Bett-Garber et al. showed that when the content of straight-chain amylose was increased, the sweetness value of the rice was reduced and the taste was not good [56]. It was hypothesized that an increase in quinoa’s straight-chain starch content would also lead to a decrease in the sweetness of cooked quinoa and, consequently, a decrease in the taste score. Subsequent determination of quinoa straight-chain starch will be carried out to verify this. Studies on rice have found that the higher the fat content within a certain range, the glossier the rice and the better the flavor [57]. However, there was no significant correlation between quinoa fat content and cooked quinoa sensory evaluation indexes in this study. Crude fiber content showed a highly significant negative correlation (*p* < 0.01) with cooked quinoa viscosity, chewiness and hardness scores, and a significant negative correlation (*p* < 0.05) with total sensory evaluation scores. The crude fiber content in quinoa is higher than that of many grains such as wheat and corn, which may result in a harder taste and rougher texture of cooked quinoa due to its higher fiber content, thus reducing the eating quality. Ash content showed a highly significant negative correlation (*p* < 0.01) with cooked quinoa chrominance and taste scores, and a significant negative correlation (*p* < 0.05) with the total sensory evaluation score. Ash is the residue of minerals in food and reflects the mineral content of the grain. Higher ash content may lead to a darker color of the quinoa, affecting its appearance and taste. In addition, changes in ash content may affect the texture and flavor of grain products [58]. Moisture content was significantly and positively correlated with cooked quinoa chrominance (*p* < 0.05), and VB1 content was significantly and positively correlated with cooked quinoa viscosity, chewiness and hardness (*p* < 0.05). The moisture content of the grains had a significant effect on their cooking time and eating quality. It has been shown that rice has the best cooking time and eating quality when stored at 14% moisture content. In addition, variation in moisture content affects the water absorption capacity and texture of grains after cooking [59]. Saponin content showed a highly significant negative correlation (*p* < 0.01) with cooked quinoa taste scores, and a significant negative correlation (*p* < 0.05) with the chrominance. Quinoa saponin had a strong bitter and astringent flavor, which was the main factor affecting the taste of quinoa rice. The low threshold of bitterness of saponins, even in small amounts, significantly affected the eating experience [60]. Saponin is water-soluble and most of them could be removed by maceration. In addition, high-temperature steaming, baking or extrusion puffing and other heat treatment methods can decompose the saponin, reduce the bitter taste, while generating new flavor substances, masking the bad taste of saponin [61].

#### 3.4.2. Principal Component Analysis and Clustering Analysis

Principal component analysis (PCA) is a statistical method that converts the original variables into linear combinations, facilitating simplification and dimensionality reduction while retaining essential information [62]. The PCA loading plot of 17 indicators for a total of 22 quinoa samples is shown in Figure 3A. The correlation between the indicators can be observed from the loading plot. The curves close to each other on the graph are positively correlated, while the curves in opposite directions are negatively correlated [63]. The curves of starch content and total score of sensory evaluation are in the same direction and close, which indicates that the sensory quality of cooked quinoa is highly positively correlated with starch content. In addition, the curves of starch content and cooked quinoa flavor index are very close to each other, indicating that the more starch cooked quinoa has, the better it tastes. Therefore, for quinoa, starch can be used as a criterion for evaluating the satisfaction of cooked quinoa. The curves for ash, protein, crude fiber and saponin content are distributed in different quadrants and in opposite directions to the cooked quinoa evaluation index curves, suggesting that the content of these components negatively affects the taste and flavor of cooked quinoa.

The distance between any two varieties of quinoa on the PCA graph refers to the similarities and differences between the quinoa. A total of 22 quinoa samples were irregularly distributed within the quadrants of the score plot based on nutrient content and sensory evaluation scores (Figure 3B). Notably, quinoa samples No. 12, No. 9, No. 17, No. 18 and No. 22 were significantly different from the remaining 17 quinoa species. Similar to the principal component score plot, cluster analysis classified the 22 quinoa species into groups based on nutrient content and sensory evaluation scores (Figure 3C). Among them, samples No. 18 and No. 22 were clearly different from the remaining quinoa varieties, with the highest crude fiber content and lower sensory index scores. Samples No. 12, No. 17 and No. 9 formed a group of their own, which had higher starch content, lower protein content, and higher scores for all sensory indexes. Consistent with the results of the electronic tongue test, samples No. 12, No. 17 and No. 9 had relatively balanced values for each taste attribute and had the best eating quality, which is in line with the pursuit of the taste of cooked quinoa.

## 4. Conclusions

There were large differences in the content of nutritional components of the 22 quinoa samples. Different nutritional components had different effects on quinoa sensory quality. In terms of chrominance, the lower the protein and ash content, and the higher the total starch and moisture content, the brighter and shinier the color of cooked quinoa. In terms of structure, the higher the content of fat, total starch and moisture, the fuller and rounder the cooked quinoa. In terms of odor, the higher the total starch content and the lower the crude fiber content, the more aromatic the quinoa cooking process. In terms of hardness, the total starch, VB1 and crude fiber content has the greatest impact. In terms of chewiness and viscosity, the lower the crude fiber content, the higher the VB1 content, and cooked quinoa will be more chewy. In terms of taste, the higher the total starch content and the lower the ash and saponin content: the cooked quinoa will taste better and be more popular. The sweetness and umami values of cooked quinoa samples No. 12 and No. 9 were relatively high, having the highest sensory scores and cooking qualities, as analyzed by electronic tongue analysis. This study preliminarily clarified the relationship between the content of quinoa nutritional components and its sensory quality characteristics, which can provide a reference for the selection of quinoa raw materials for different consumption methods and variety selection. In the future, we will conduct more detailed studies on the processing quality of quinoa to provide more data support for more effective improvement of quinoa processing quality.

## Figures and Tables

**Figure 1 foods-14-00988-f001:**
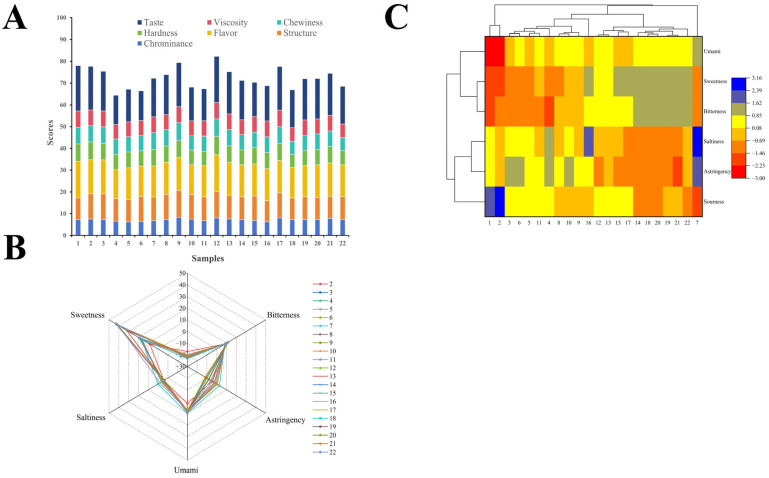
Sensory evaluation score (**A**), radar plots (**B**) and cluster heat map (**C**) of electronic tongue sensor response of cooked quinoa samples.

**Figure 2 foods-14-00988-f002:**
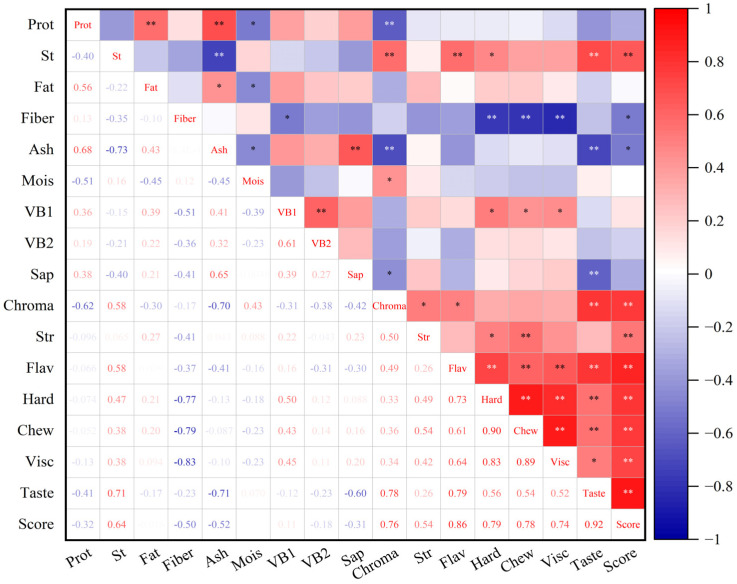
Correlation analysis based on nutritional quality and sensory evaluation of quinoa samples. * Represents a significant difference between data, *p* < 0.05, ** represents an extremely significant difference between data, *p* < 0.01. There are the full names and abbreviations: Protein (Prot), Starch (St), Moisture (Mois), Saponin (Sap), Chrominance (Chroma), Structure (Str), Flavor (Flav), Hardness (Hard), Chewiness (Chew), Viscosity (Visc).

**Figure 3 foods-14-00988-f003:**
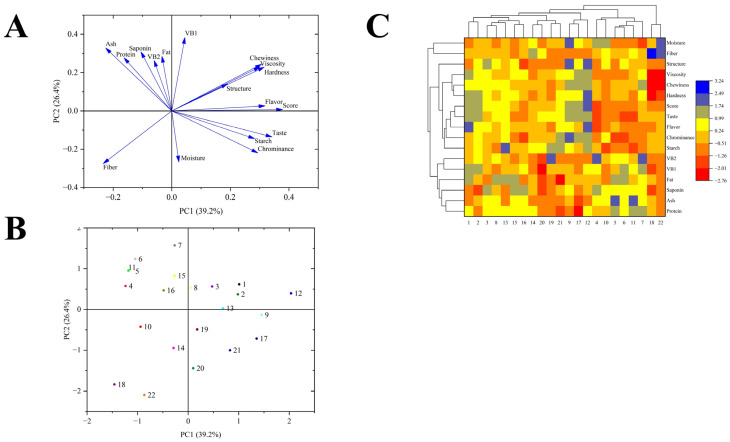
PCA analysis and cluster analysis based on nutritional quality and sensory evaluation of quinoa samples. The loading plot (**A**), score plot (**B**) and cluster heat map (**C**). Different color dots in score plot (**B**) represent different samples.

**Table 1 foods-14-00988-t001:** Information on the 22 quinoa samples.

Sample Number	Variety	Color	Origin
1	Suli 1	Light brown	Jiangsu
2	Suli 2	White	Jiangsu
3	Mengli	White	Inner Mongolia
4	Chuangli 1	Dark brown	Sichuan
5	Liangli 1	Dark brown	Sichuan
6	Jinli 1	Light brown	Shanxi
7	Jinli 2	Light brown	Shanxi
8	Jinli 3	Black	Shanxi
9	Jili 3	White	Hebei
10	Jili 5	Black	Hebei
11	Longli 1	Light brown	Gansu
12	Qinghai white quinoa	White	Qinghai
13	Qinghai red quinoa	Red	Qinghai
14	Qinghai black quinoa	Black	Qinghai
15	Yili 1	Light brown	Xinjiang
16	Yili 2	Light brown	Xinjiang
17	Peruvian white quinoa	White	Peru
18	Peruvian blacke quinoa	Black	Peru
19	Bolivian red quinoa	Red	Bolivia
20	Bolivian blacke quinoa	Black	Bolivia
21	Bolivian white quinoa	White	Bolivia
22	Yunnan black quinoa	Black	Yunnan

**Table 2 foods-14-00988-t002:** Nutrient content of quinoa samples.

Sample	Starch (g/100 g)	Protein (g/100 g)	Fat (g/100 g)	Fiber (g/100 g)	Ash (g/100 g)	VB1 (mg/100 g)	VB2 (mg/100 g)	Moisture (%)	Saponin (mg/g)
1	53.8 ± 0.97 def	15.21 ± 0.34 efg	8.11 ± 0.12 c	2.02 ± 0.06 k	2.30 ± 0.01 j	0.41 ± 0.02 c	0.28 ± 0.01 g	9.89 ± 0.07 op	1.79 ± 0.02 l
2	56.37 ± 1.50 bcde	13.09 ± 0.41 i	6.59 ± 0.06 fg	2.68 ± 0.03 h	2.90 ± 0.01 g	0.45 ± 0.01 a	0.32 ± 0.01 d	10.28 ± 0.02 kl	1.45 ± 0.01 n
3	52.5 ± 1.31 efg	14.92 ± 0.00 fg	5.62 ± 0.05 l	1.82 ± 0.03 l	3.21 ± 0.02 f	0.32 ± 0.01 g	0.23 ± 0.00 jk	10.50 ± 0.08 j	2.75 ± 0.03 c
4	50.62 ± 0.50 efg	13.92 ± 0.04 h	6.35 ± 0.11 h	2.16 ± 0.05 j	3.19 ± 0.04 f	0.39 ± 0.00 d	0.40 ± 0.02 c	11.91 ± 0.12 d	3.03 ± 0.04 a
5	47.94 ± 0.86 fgh	16.28 ± 0.32 bc	7.93 ± 0.03 d	2.80 ± 0.01 g	4.19 ± 0.03 b	0.32 ± 0.00 g	0.29 ± 0.00 fg	10.02 ± 0.07 mn	2.46 ± 0.02 f
6	47.27 ± 0.35 fgh	15.63 ± 0.32 d	8.43 ± 0.08 b	2.27 ± 0.04 i	3.39 ± 0.04 de	0.46 ± 0.01 a	0.30 ± 0.01 ef	9.58 ± 0.04 q	2.54 ± 0.03 e
7	46.31 ± 0.72 gh	16.79 ± 0.08 a	7.28 ± 0.15 e	2.14 ± 0.07 j	3.50 ± 0.07 c	0.43 ± 0.01 b	0.42 ± 0.02 b	9.42 ± 0.02 r	2.47 ± 0.01 f
8	50.4 ± 0.63 efg	15.29 ± 0.01 def	7.94 ± 0.10 d	2.31 ± 0.02 i	3.23 ± 0.04 f	0.26 ± 0.00 j	0.31 ± 0.01 de	9.80 ± 0.02 p	2.07 ± 0.03 h
9	56.55 ± 1.03 bcde	12.11 ± 0.05 k	5.98 ± 0.04 j	0.86 ± 0.03 p	2.45 ± 0.06 i	0.30 ± 0.01 h	0.21 ± 0.00 l	12.57 ± 0.02 b	2.54 ± 0.04 e
10	45.3 ± 1.03 i	14.07 ± 0.09 h	5.53 ± 0.05 l	3.34 ± 0.02 e	3.34 ± 0.04 e	0.22 ± 0.00 k	0.28 ± 0.01 g	12.14 ± 0.06 c	2.61 ± 0.02 d
11	42.47 ± 0.44 h	16.49 ± 0.00 ab	6.7 ± 0.03 f	3.27 ± 0.03 f	4.35 ± 0.03 a	0.40 ± 0.02 cd	0.25 ± 0.01 hi	9.95 ± 0.02 no	2.64 ± 0.04 d
12	62.94 ± 1.13 ab	14.79 ± 0.16 g	6.49 ± 0.09 g	0.76 ± 0.01 q	1.90 ± 0.02 m	0.37 ± 0.01 e	0.21 ± 0.00 l	10.82 ± 0.02 h	2.27 ± 0.02 g
13	64.03 ± 1.30 a	16.03 ± 0.38 c	8.88 ± 0.06 a	1.86 ± 0.03 l	2.23 ± 0.02 k	0.29 ± 0.00 hi	0.24 ± 0.00 ij	10.69 ± 0.04 i	1.82 ± 0.02 kl
14	55.23 ± 1.65 cde	15.36 ± 0.27 de	5.77 ± 0.16 k	4.72 ± 0.02 c	2.09 ± 0.07 l	0.23 ± 0.00 k	0.21 ± 0.01 l	11.59 ± 0.09 e	2.01 ± 0.03 i
15	54.21 ± 0.94 def	14.79 ± 0.35 g	8.77 ± 0.13 a	1.66 ± 0.03 m	3.45 ± 0.04 cd	0.28 ± 0.01 i	0.21 ± 0.00 l	10.10 ± 0.04 m	2.98 ± 0.04 b
16	53.78 ± 1.86 def	15.08 ± 0.28 efg	4.8 ± 0.04 o	1.85 ± 0.01 l	3.20 ± 0.02 f	0.35 ± 0.00 f	0.26 ± 0.01 h	10.91 ± 0.03 gh	2.71 ± 0.02 c
17	54.45 ± 1.89 cdef	8.99 ± 0.01 m	6.2 ± 0.08 i	1.22 ± 0.04 no	1.47 ± 0.04 p	0.25 ± 0.00 j	0.22 ± 0.01 kl	11.43 ± 0.10 f	1.82 ± 0.01 kl
18	50.66 ± 0.75 efg	14.29 ± 0.12 h	6.22 ± 0.07 hi	8.67 ± 0.02 a	2.62 ± 0.03 h	0.16 ± 0.00 m	0.17 ± 0.00 m	10.36 ± 0.02 k	1.48 ± 0.03 n
19	60.09 ± 0.93 abcd	11.96 ± 0.12 kl	5.16 ± 0.13 n	1.25 ± 0.04 n	1.87 ± 0.06 m	0.32 ± 0.01 g	0.44 ± 0.01 a	11.00 ± 0.05 g	1.84 ± 0.02 k
20	56.99 ± 1.16 bcde	12.62 ± 0.29 j	4.19 ± 0.04 p	3.70 ± 0.02 d	2.09 ± 0.03 l	0.12 ± 0.00 n	0.16 ± 0.01 m	10.20 ± 0.02 l	1.45 ± 0.02 n
21	61.19 ± 0.66 abc	11.61 ± 0.12 l	2.94 ± 0.03 q	1.18 ± 0.03 o	1.80 ± 0.05 n	0.29 ± 0.00 hi	0.21 ± 0.01 l	10.82 ± 0.07 h	1.97 ± 0.03 j
22	50.64 ± 1.41 efg	13.02 ± 0.37 i	5.38 ± 0.11 m	6.24 ± 0.01 b	1.73 ± 0.04 o	0.20 ± 0.00 l	0.17 ± 0.00 m	12.99 ± 0.09 a	1.59 ± 0.01 m
Average	53.35	14.20	6.42	2.67	2.75	0.31	0.26	10.77	2.19
Standard deviation	5.70	1.90	1.52	1.86	0.81	0.09	0.08	0.97	0.50
Coefficient of variation (%)	10.68	13.37	23.65	69.50	29.51	29.83	29.72	9.05	22.90

The data were based on dry weight and expressed as mean ± standard deviation (SD) of the measured values (*n* = 3). Different letters indicate significant differences (*p* < 0.05).

## Data Availability

The original contributions presented in the study are included in the article, further inquiries can be directed to the corresponding author.

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
