# Peer review of "Influence of the Nutritional Composition of Quinoa (Chenopodium quinoa Willd.) on the Sensory Quality of Cooked Quinoa"

_foods, 2025, doi:10.3390/foods14060988_

Round 1
Reviewer 1 Report
Comments and Suggestions for Authors
Dear Authors,
COMMENTS- foods-3430307
The manuscript "Correlation analysis between nutritional components and edible quality of quinoa" reconnoiters and most demanding issue for the popularization of the time-honored nutritious crop, quinoa. The author's efforts on this underutilized nutritious crop are appreciable, they explore the nutritional components of quinoa from 22 varieties collected from 11 major quinoa-producing areas. The manuscript provides an overview of the relationship between the content of quinoa nutritional components and the edible quality characteristics, which can provide a future way to popularize this crop as a daily diet component on a mass level. However, the manuscript presented nicely with support of tables, and figures but critical methodological shortcomings that significantly affect on recommendation of the findings. The following sections are missing in the manuscript:
1. Add saponin study – Saponins are an important group found in Chenopodium quinoa prominently and represent an obstacle to the use of quinoa as food because of their bitter taste and toxic effects, they can negatively impact the nutritional value of food by interfering with the absorption of certain nutrients like minerals (iron, zinc, calcium) and vitamins (A, E), primarily due to their ability to bind to these nutrients in the digestive tract, potentially reducing their bioavailability and leading to decreased nutrient uptake by the body. It is compulsory to add this important aspect to recommend quinoa varieties with lower saponin and higher nutritional and edible qualities.
2. Absence of bioavailability of nutrients— This absence is concerning, as the choice of quinoa varieties with better nutrition for the human body, at present the experimental recommendation based on the available nutrition of quinoa seed.
3. Digestibility of starch and protein — Without the determination of the starch and protein digestibility the experimental recommendation is unconcluded, it is suggested to observe the in vitro protein digestibility (%) and in vitro starch digestibility (%) for better recommendation.
4. Statistical upgradation- Statistically significant means were compared with the Tukey multiple comparison tests required for the evaluation of nutrient content variation in quinoa seed in Table 2.
The following upgradation also required-
1. Research Title: The title of the manuscript is informative and appropriate; it requires the following correction.
“Correlation analysis between nutritional components and edible quality of quinoa
(Chenopodium quinoa Willd.)”
2. Abstract: The abstract part is nicely presented. If possible, also add saponin and digestibility components.
3. Introduction: This section requires some corrections as per the enclosed PDF file. Also, add some facts about quinoa production, consumption, and major producing countries with consumable product details.
4. Materials and Methods: It is an insufficient, upgradation required.
i) Table 1 adds some important phenotypic characters related to the nutritional quality of seeds, such as seed colour, seed test weight, seed density, seed hardness, etc. Also, add Saponin contained in all 22 varieties of seeds used in experimentation.
ii). Process of saponin, in vitro digestibility of starch and protein
5. Results and Discussion: The results of the experiment are properly presented with the help of tables. However, the discussion part is poorly written it requires logical factors to justify the experimental outcomes with closely related previous study supports. Also, incorporate the following suggestions. See PDF file
i). Apply Tukey multiple comparison tests on the table no 2.
ii). Delete or rewrite or refer properly to the sentence from lines 184 to 190, concerning human studies such as the weight control effect.
iii). Add results of Fibre contain, VB1 and VB2 in the nutrient contain section
iv). Restructure the lines 144 to 145.
v). Add discussion with appropriate references in the sensory evaluation result section.
vi). Add discussion with appropriate references in the electronic tongue analysis results section.
6. Conclusion: Nicely presented.
7. Tables: In both tables add required information and rearrange.
9. Reference: Presented as per journal guidelines, only highlighted references required formatting.
13. English language: The English language of the manuscript is nice.

The English language of the manuscript is nice
Author Response
Many thanks for the helpful comments and thoughtful suggestions from you. Based on these comments and suggestions, we have made careful modifications to the original manuscript. Please see the attachment.

Reviewer 2 Report
Comments and Suggestions for Authors
The title of the work corresponds to the subject of conducted and described research.
The introduction provides a comprehensive implementing to the subject matter and justifies the purpose of the research undertaken.
The research material deserves special attention. There have been 22 varieties of quinoa from 11 regions of its cultivation collected. There are few works conducted on such extensive material.
Research methods
In line 102, the abbreviations VB1 and VB2 should be expanded. Using these abbreviations without explanation may not be understandable for all readers.
2.4. Sensory evaluation of cooked quinoa It is difficult to find the sensory evaluation methodology (literature item unavailable across the internet). Were the evaluators trained? Has the consent been obtained for a sensory evaluation?
2.5. Electronic tongue analysis - In which units were the tested features evaluated?
Results and discussion
Lines 142-143 - Does the sentence “The amylose content of quinoa varieties from the same province was similar, and the amylose content of quinoa grown in different provinces had significant differences.” refer to your own research or any literature data? In the studies, the starch content was determined, not the amylose content. However, if the above sentence concerns any specific literature data, the appropriate authors should be cited. Moreover, why has the significance of differences in the content of individual components not been assessed using statistical methods? It is necessary to perform calculations of the coefficient of variation, or even better, ANOVA variance analysis so that it is possible to determine the significance of the differentiation with a specified probability and designate homogeneous groups, and in chapter 2.6. Statistical analysis - list these calculation methods.
Line 155 - It is not true that "protein content of sample No. 7 (16.79%) was significantly higher than that of other quinoa". Samples 5 (16.28 g/100g), 11 (16.49 g/100g) and 13 (16.03 g/100g) also had high protein content similar to that of sample No. 7.
Lines 163 -164 - The authors claim - "and there have been many reports on the effect of protein on the edible quality of quinoa", and then they refer to only one literature item by Wu et al. More publications works need to be cited.
Line 172-173 The authors found that the coefficient of variation calculated for fat content is the highest of all those calculated for the other components., and such values ​​were not given in the work and probably were not calculated because this method of statistical processing of results was not mentioned in chapter 2.6.
Lines 183-192 The discussion of the content of such components as dietary fiber, ash and moisture is too short. There is almost no discussion of the content of vitamins VB1 and VB2, because the statement that "Quinoa is rich in water-soluble B vitamins and is a good natural source of vitamins." is very general and may result from some literature data, which the Authors do not refer to.
The description of the results presented in Table 2 is not very insightful. Having such a rich experimental material, more could be deduced from the results obtained than that sample no. ... has the most component, and sample no. ... has the least. Different varieties from different regions were tested, and in the discussion this was completely ignored. Such conclusions would be very interesting
Line 199 It is worth to include the coefficients of variation listed in this line in Figure 1A, e.g. in the legend - in brackets next to the names of the features determined in the sensory evaluation.
Lines 209-210 - From Figure 1A it is clear that not only Samples 4 and 6 do have a low overall rating, Sample 18 appears to have a similar rating to Sample 6.
Lines 219-220 Note to the sentence “There was a significant difference (p<0.05) between the values ​​of the six taste attributes of different varieties of cooked quinoa.” – The significance of the differences in the results obtained for the 22 samples can be estimated for each of the 6 taste attributes, not the values ​​of the six taste attributes. Furthermore, the results shown in Figure 1B indicate that sweetness and astringency are more diverse than the others. It would be worthwhile to indicate those samples that were characterized by better sensory properties.
Lines 232 and 234 The authors refer to the freshness values ​​of samples No. 7, 9 and 12 – the paper does not present freshness results, so where does this description come from?
Lines 239-270 of chapter 3.4.1. Correlation analysis – no possibility to check the correctness of the interpretation of results due to the complete illegibility of Figure 2A
Conclusions
With the exception of the first sentence, which is not supported by the presented results (lack of statistical evaluation of the variability of the chemical composition of the tested samples), the conclusions correspond to the obtained results.
Table and Figures
In Table 2, if the place of origin of the samples were inscribed, differences due to environmental conditions would be more pronounced. Furthermore, it lacs coefficients of variation or definition of homogeneous groups using ANOVA analysis.
Figure 1 and 2 are difficult to read - first of all, the font size on the graphs should be increased
Figure 1C and 2D - what do the numbers in the figure legend mean? This is not described in the Research Methods chapter or, neither in the figure at all.
Figure 2A - The results and descriptions of the axes presented in this figure are completely illegible even at 300% magnification
Author Response

(The authors gave the same response as above.)
